# Influence of Cross-Grooved Texture Shape on Tribological Performance under Mixed Lubrication

**Song Hu [1,2], Long Zheng [1,2,*], Qinggang Guo [1,2] and Luquan Ren [1,2]**

1    Key Laboratory of Bionic Engineering (Ministry of Education), Jilin University, Changchun 130025, China; 18332580119@163.com (S.H.); guoqg1997@163.com (Q.G.); lqren@jlu.edu.cn (L.R.)
2    Weihai Institute for Bionics, Jilin University, Weihai 264402, China
*    Correspondence: zhenglongcclg@163.com

**Abstract:** Surface texture plays an important role in improving the tribological properties of materials. In this paper, the effect of different shapes (i.e., triangle, square, hexagon, round) on the tribological performance of cross-grooved texture was investigated. First, the mixed lubrication condition was used for the pin-on-disc rotating sliding tests. Then, the stress distribution of the four textures was analyzed to better explain the experimental results. Overall, the hexagon-textured specimens exhibited lower friction coefficients than the other shape-textured specimens under the examined conditions. Simulation results indicate that the contact stress can be reduced on the surface of hexagon-textured specimens, and this leads to a better oil film for lubrication. Furthermore, the hydrodynamic lubrication stood out with the increase of speeds to 250 rpm. However, as the test loads further increased, the film thickness decreased, resulting in the increase in the asperity contact areas, which dropped the above advantage of hexagon-textured specimens. This study would be beneficial for the texturing tribological and lubrication design.

**Keywords:** cross-grooved; texture shape; tribological performance; mixed lubrication

## 1. Introduction

Friction has always been the major concern of part failure, which causes an excessive loss of resources. Surface texturing, as an effective engineering technology to improve the anti-wear properties, has been proposed and studied by many scholars [1–3]. Surface textures have been widely investigated in recent decades and proven to be effective at reducing friction force and improving wear resistance under normal workpiece conditions with better tribological properties [4–7]. Furthermore, surface texturing has been successfully applied to sliding bearings, machine components in sliding contacts, cylinders of internal combustion engines, and mechanical seals [8–10]. Table 1 summarizes the recent research on surface texture for practical engineering applications.

The generally accepted mechanisms to explain the friction reduction by surface textures include textures that can store lubricant to introduce secondary lubrication and entrap wear debris or impurities to protect the contact surface from scratching again [16–18]. Meanwhile, the load carrying capacity of hydrodynamic lubrication is increased due to the cavitation effect.

The design of new texture shape is an effective way to further improve the tribological performance. The shape form of surface texturing is divided into two general types: a discrete (dimple) texture and a continuous one (groove) [19–21]. The dimple, as the general texture form, is the most investigated, and the tribological properties of various dimple shapes have been studied extensively. The oil pockets of spherical and drop shapes textured on the steel rings have been tested under lubrication; the results of their abrasive wear demonstrated that the dimple of the spherical shape was superior to that of the drop shape in terms of wear resistance [22]. In addition, the ring surfaces with oil pockets of short drop, long drop, and spherical shapes were compared to clarify the influence of the geometrical

characteristics of the surface texture on the Stribeck curve in lubricating sliding; the results showed that the proper textured shape can improve lubricating characteristics [23]. As such, the role of round dimple, diamond dimple, and ellipse dimple were distinguished under boundary and mixed lubrication, during which the round dimple geometry yielded the lowest friction and wear among others [24]. In a recent study, the numerical simulation can also predict the effect of texture shapes on surface tribological properties: the influence of six types of pit shapes on the friction coefficients of gas-lubricated parallel slider bearings (i.e., circle, ellipse, sphere, ellipsoid, triangle, and chevron) were reconstructed by numerical simulation, and it drew the conclusion that the ellipsoidal shape obtained the lowest friction coefficient and the highest bearing stiffness [25]. Similarly, another numerical simulation based on the sequential quadratic programming algorithm was used to optimize texture shapes for lubrication and acquired the results that chevron-shapes and trapezoid-like shapes generated the load carrying capacity (LCC) in the corresponding sliding directions and always had a greater LCC at the area ratio of 30% [26]. Moreover, the geometric shape (i.e., ellipse dimple, circle dimple, and triangle dimple) effect of the textured surface on hydrodynamic pressure were investigated by both experiments and theoretical models. It was found that the geometric shapes of dimples had a non-negligible influence on the LCC, and the textured specimens with ellipses perpendicular to the direction of sliding showed the highest LCC [27]. In order to maximize the LCC of the textured surface, the level set method was introduced to optimize the shape of the surface texture with dimples under the cavitating hydrodynamic lubrication condition. The results showed that the optimal geometries were the chevron-type shape [28].

**Table 1.** Application of surface texture in practical engineering.

| Application | Texture | Results |
| --- | --- | --- |
| Cutting tool | Vertical groove texture [11] | Reduce friction by 11.9% |
| Piston ring | Partially texture with 'open pockets' [12] | The total friction dropped by 15% |
| Cylinder liner | Honing & circular texture [13] | Peak engine power increased by 5.8% |
| Mechanical seals | Asymmetric 'V' shapes texture [14] | Provided higher load-carrying capacity than conventional shapes |
| Journal bearing | Dimple texture [15] | The friction is reduced by up to 18% |

Correspondingly, different forms of continuous grooves were also considered as an alternative option for friction reduction. The parallel grooves and crossed grooves were compared at high pressure and low sliding speed under lubrication, and the effect of the crossed grooves on the reduction in friction coefficients relative to parallel grooves stood out [29]. Other studies found that wavy grooves led to a lower friction coefficient in both unlubricated and MoS2-lubricated conditions than parallel grooves [30]. Additionally, textured specimens with grooves, circle dimples, and chevron-like dimples at different orientations, depths, densities, and aspect ratios were tested under hydrodynamic lubrication. Among all these texturing features, circle dimples and chevron-like dimples had better tribological properties by generating hydrodynamic films with the highest thickness [31]. When grooves were applied on the surface of non-metallic materials (e.g., a poly(dimethylsiloxane) (PDMS) elastomer), pillar-textured specimens could also minimize the friction coefficients by 59% compared with non-textured specimens [32]. Inspired by microstructures on the animal surface, recently, the bionic texture (e.g., hexagonal texture) has been proposed by some researchers and its superiority in reducing friction has been verified experimentally and through modeling [33,34].

However, except for the mentioned parallel or other continuous grooves, there is little research into how the cross-grooved texture shapes tune the tribological properties. The cross-grooved texture is composed of a pattern of interlacing repeated units, which include square, triangle, diamond, hexagon, etc. While only triangles, squares, and hexagons can cover the surface effectively, hexagonal patterns are more frequently used in nature (e.g., within beehives), and are the most effective way to pack the largest number of

similar objects in a minimum space [35,36]. Inspired by this fact, four types of cross-grooved texture shapes (i.e., triangle, square, hexagon, round) have been fabricated on AISI 1045 steel specimens by a fiber laser to compare the friction coefficients with the change in loads and rotating speeds under mixed lubrication.

## 2. Materials and Methods

### 2.1. Fabrication of Specimens

The samples were cut from AISI 1045 steel into small cylinders with a diameter of 35 mm and a height of 16 mm. The cylindrical counter-disks with a diameter of 80 mm and a height of 14 mm in the tribo-testing were prepared from AISI 52100 steel. The traditional heat treatment was performed on the samples and counter-disks to enhance the hardness of the surface, and the sequence of the heat treatment process for AISI 1045 steel is annealing, quenching, and tempering. The Rockwell hardness of specimens and counter-disks were 48 and 60 HRC, respectively.

Then, laser surface texturing (LST) was used to engrave four shape textures on the surface of the sample. Among all the texture-manufacturing technologies, LST was chosen because of its characteristics of simple operation, high efficiency, and low pollution [37,38]. A fiber laser with a 1064 nm wavelength was used. The laser processing used the laser power of 30 W with a scanning speed of 500 mm/s. The defocused laser was 0.05 mm in diameter, and the pulse duration was 20 kHz [39]. Both the specimens and counter-disks were polished by abrasive paper and grinding paste to achieve the RMS surface roughness (Ra) of 0.2 μm after laser texturing [33].

The schematic diagram of the specimen is shown in Figure 1. In order to explore the effect of cross-grooved texture shapes on the lubricated characteristics, all the areas of simple shapes were the same and the groove width was 0.5 mm. The specimens were replicated. The micrographs of the textured groove were obtained by the principles of White Light Interferometry, and the groove depth was measured to be 10 μm, as shown in Figure 2. For a better comparison, the preparation process of the non-textured specimen was the same, only without laser processing. Before each test, the specimens and counter-disks were cleaned with a mixture of acetone (5%) and ethanol (95%) in an ultrasonic apparatus.

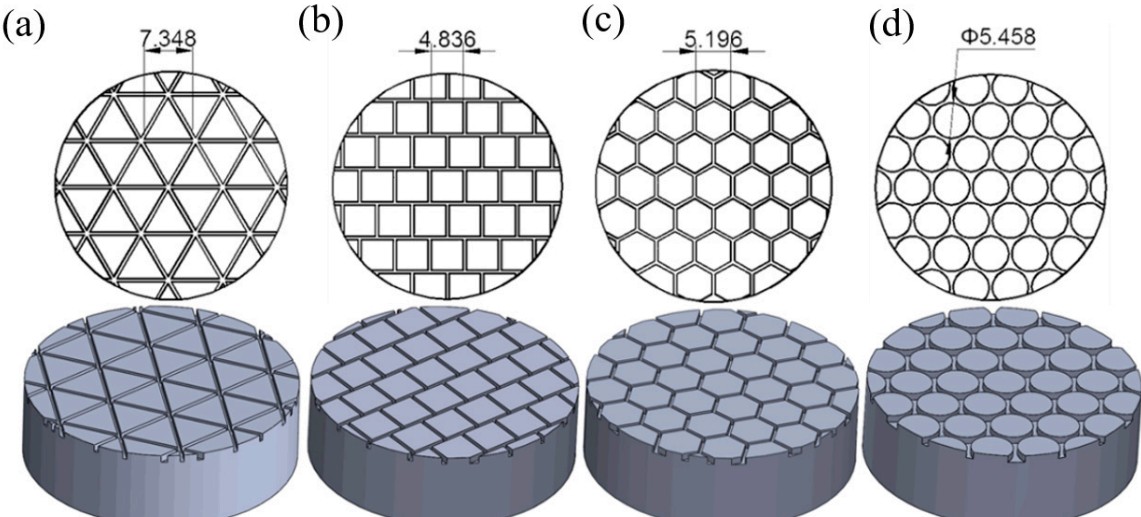

**Figure 1.** The schematic diagram of the specimens with four types of cross-grooved texture shapes: (**a**) triangle-textured specimen; (**b**) square-textured specimen; (**c**) hexagon-textured specimen; and (**d**) round-textured specimen.

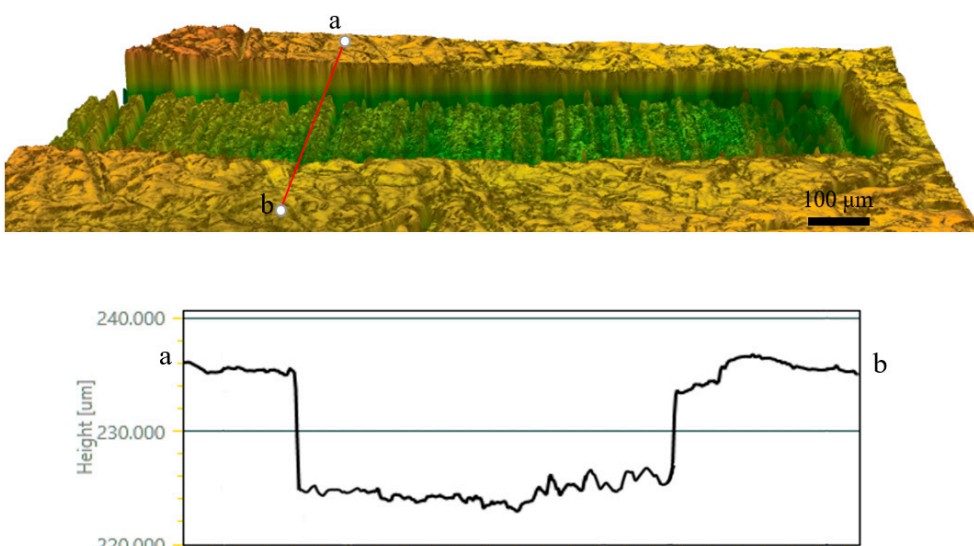

**Figure 2.** Micrographs of the textured surface. The average groove depth was about 10 μm.

### 2.2. Tribo-Testing

The friction tests under lubrication were implemented at ambient temperature and humidity (around 28 °C and 30% RH) with a tribo-testing machine (M2000, Hebei Testing Machine Co. Ltd., Hebei, China) using a disc-on-pin setup shown in Figure 3. The upper was the counter-discs, and the lower was the tested specimen, the lubrication of which could flow through the grooves. A commercial diesel (0#) with a viscosity (η) of 0.0027 Pa·s was selected as the lubricant, and 0.6 mL of lubricant was applied in each test. The tests were conducted under a series of loads (20, 50, 70, and 100 N) and rotating speeds (50, 100, 200, and 250 rpm). The testing period was 60 min, after which the friction coefficient was recorded. Five specimens were repeated for each condition to reduce test errors.

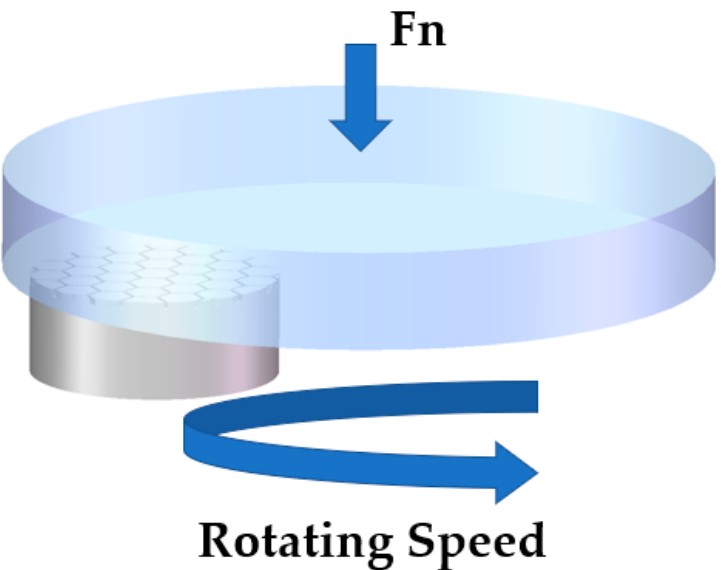

**Figure 3.** The schematic diagram of the frictional test equipment.

### 3. Results and Discussion

Textured specimens with the four types of shapes were conducted by tribo-testing under lubrication conditions and untextured specimens were compared. The average friction coefficients of two independent experiments under the selected loads and rotating

speeds were compared, as shown in Figure 4. Overall, all the specimens exhibited high friction coefficients under the load of 20 N, and the friction coefficients would reduce with the increase in loads. Furthermore, the friction coefficient (COF) of the textured specimens were generally lower than that of the untextured samples, although the coefficient of friction was comparable in a few cases. This observation was consistent with the findings in previous studies [1,10].

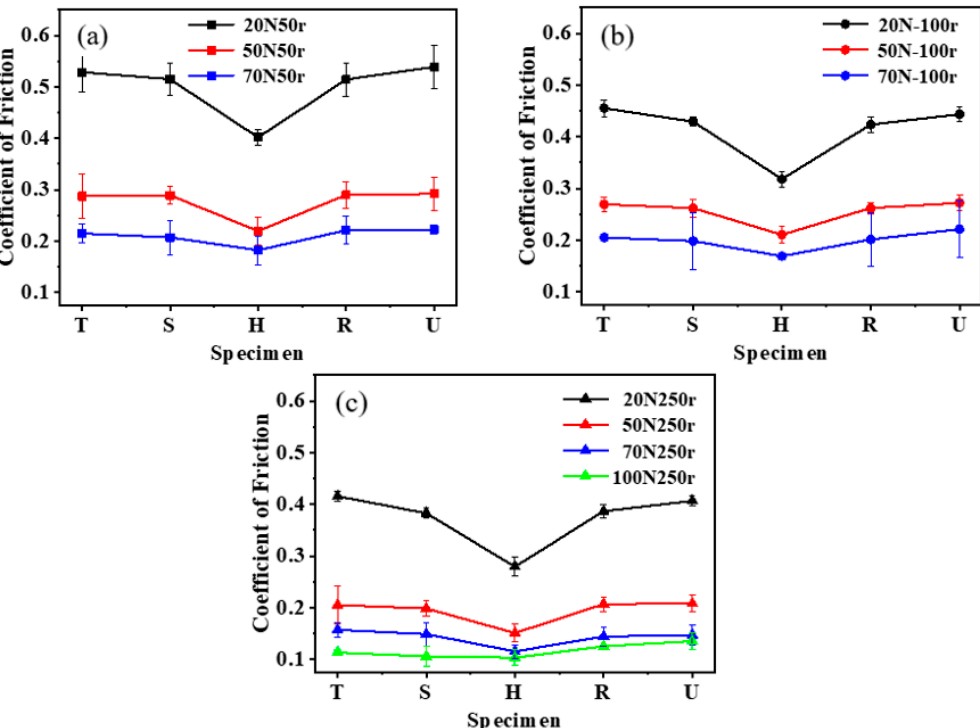

**Figure 4.** The average friction coefficients of four cross-grooved texture shapes under the rotating speeds of (**a**) 50 r, (**b**) 100 r, and (**c**) 250 r. T (triangular); S (square); H (hexagonal); R (round); and U (untextured).

Interestingly, the hexagon-textured specimens showed the lowest friction coefficients compared to textured specimens with other shapes. For the minimum loads of 20 N (the black polyline in Figure 4), there was an obviously significant difference in COF between hexagon-textured specimens and other shape-textured specimens; the difference of them mitigated as the loads increased. When the load reached 70 N, with a rotating speed of 250 rpm (the blue polylines in Figure 4c), the difference between hexagon-textured specimens and other shape-textured specimens was still obvious. As the load further increased to 100 N, the difference faded away (the green polyline in Figure 4c).

For a better visualization of the above trends, the relative increases in COF (COF$_{ri}$) are shown in Figure 5. COF$_{ri}$ is defined as follows:

$$COF_{ri} = \frac{COF_T, \ COF_S \text{ or } COF_R - COF_H}{COF_H} \tag{1}$$

COF$_T$, COF$_S$, COF$_H$, and COF$_R$ are defined as the COF of triangle-textured, square-textured, hexagon-textured, and round-textured specimens, respectively. It is evident that the COF of the hexagon-textured specimens showed the greatest improvement at the minimum load of 20 N, but the friction coefficient improvement became less clear at the higher loads of 70 N or 100 N. Particularly, the hexagon-textured specimens also exhibited the lowest COF compared to other shape-textured specimens under different rotating speeds, and this advantage for hexagon-textured specimens in lubrication increased with the rotating speed in Figure 5.

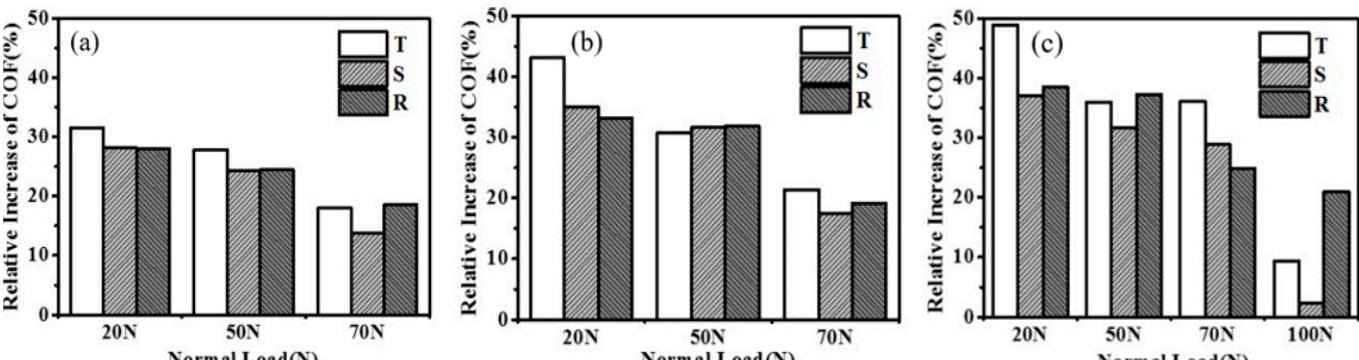

**Figure 5.** The influence of the other three textured specimens on the relative increases in COF compared with that of hexagon-textured specimens under different rotating speeds: (**a**) 50 rpm; (**b**) 100 rpm; (**a**) 250 rpm.

Figure 6 shows the interactions between the friction coefficient and the rotating speeds for four species of textured-specimens. It can be seen that all the specimens yielded high friction coefficients under the rotating speeds of 50 rpm, and then the friction coefficients decreased with the increase in rotating speeds. This indicates that the hydrodynamic lubrication is dominant and the oil film has a load-carrying capacity, which is in accordance with former studies [40,41]. Excitingly, the hexagon-textured specimens also exhibited the lowest COF compared to other shape-textured specimens under different rotating speeds. Moreover, the advantages in the lubrication of hexagon-textured specimens increased with the rotating speed, as shown in Figure 7. As the rotating speeds increased, the difference between the hexagon-textured specimens and other shape-textured specimens gradually increased.

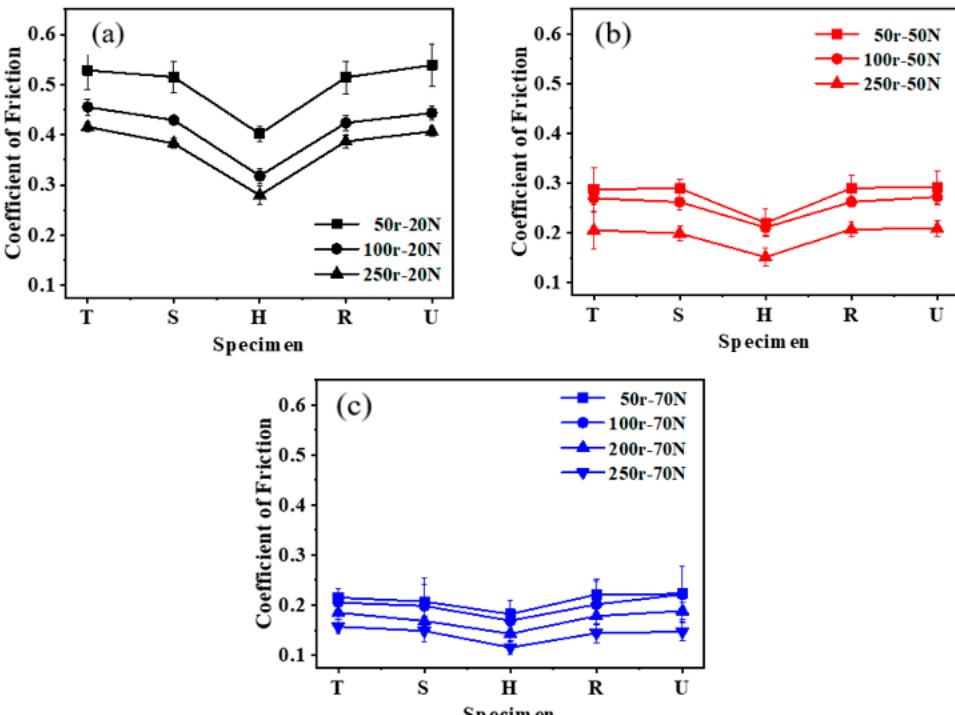

**Figure 6.** The average friction coefficients of four cross-grooved texture shapes under the loads of (**a**) 20 N, (**b**) 50 N, and (**c**) 70 N.

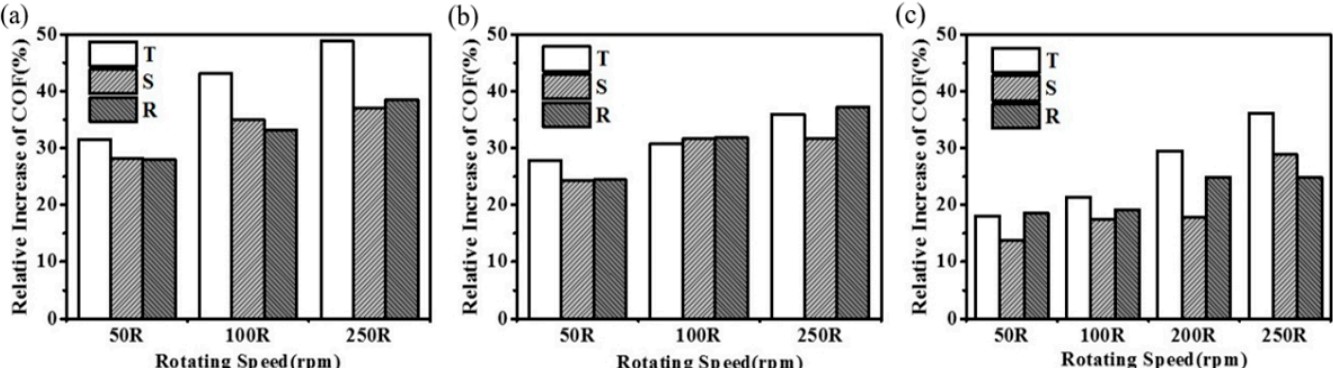

**Figure 7.** The influence of the other three textured specimens on the relative increases of COF compared with that of hexagon-textured specimens under different loads: (**a**) 20 N; (**b**) 50 N; (**a**) 70 N.

According to the experimental conditions and the obtained friction coefficients, it can be presumed that the specimens are mainly conducted under mixed lubrication. In the mixed lubrication, the asperity contacts and fluid film play an important role in load capacity [42]. When the load is constant, the larger the contact area, the less contact pressure per unit area. Meanwhile, high contact pressure appears at the edge of the grooves and become adverse to lubrication by increasing the probability of asperity contact [43]. Although the four textured specimens had the same areas, their area densities (calculated as the ratio of the total area excluding the texture to the whole area of the specimen) were 77.82% for triangular texture, 79.37% for square texture, 84.00% for hexagonal texture, and 77.32% for circular texture, respectively. Meanwhile, assuming that the oil volume on the surface is not changed, since the groove can achieve the function of oil storage, the groove will lead to a decrease in the thickness of the oil film on the surface, thereby decreasing the distance of contact surface and asperity contacts. Hexagon-textured specimens have the largest area densities compared to other shape-textured specimens, which imposes less contact pressure on the surface of hexagon-textured specimens and weakens the acting force of asperity contact. This may be one of the reasons for the reduction in the friction coefficient. Furthermore, the oil volume on the surface of hexagon-textured specimens was more than the others because of the smallest number of grooves, which separates the upper and lower surface in favor of lubrication in the clearance between contact surfaces. Unfortunately, as the loads increase, the above performance is not prominent due to the limited carrying capacity and the increased asperity contact areas. In addition, the oil between the contact areas will be squeezed because of the higher loads, so the difference in COF of each specimen-textured with each shape is not as obvious. Therefore, it is believed that the hexagon-textured specimens have a remarkable load capacity at the load of 20 N.

In order to better understand the influence of the oil film on the load-carrying capacity of the four texture specimens, a simulation model was set up in the commercial software ANSYS. To simplify the model, each groove shape unit able to cover the surface of specimens in the *X* and *Y* axis of cartesian coordinate system was selected for study, as shown in Figure 8.

The simulation experiment in this part was completed in ANSYS Workbench. Apply a series of loads (20, 50, 70, and 100 N) to the textured surface. The mesh was generated by tetrahedral elements, the element size was 3 mm. The mesh on the surface of the tested specimen was refined with an element size of 0.2 mm. Figure 9 shows the meshed hexagonal texture specimen. Other options used default values. By applying the analysis over the texture surface under load, we obtained the stress distribution numerically. Figure 10 shows the simulation results of the equivalent stress on the surface of each texture-unit. It can be seen that the hexagonal-unit exhibits uniform stress distribution and less equivalent stress than other textured-units, except for the triangle-unit. We should note that the maximum equivalent stress of hexagonal-unit was nearly equal to that of the triangle-unit (3.7752 and 3.5441, respectively), but the area of the triangle-unit was twice that of the

hexagonal-unit. However, the edge loading makes peak values potentially misleading. Overall, the equivalent stress was concentrated at the edge, especially the round-unit. It can be seen from the round-unit that the equivalent stress in a fair portion of the region (dark blue) was almost zero, which caused the greater stress at the edge.

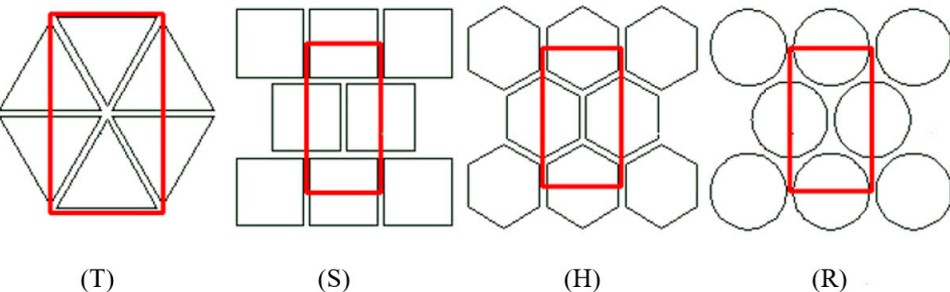

(T)        (S)        (H)        (R)

**Figure 8.** The simulation units of four cross-grooved texture: (T) triangle-textured; (S) square-textured; (H) hexagon-textured; (R) round-textured.

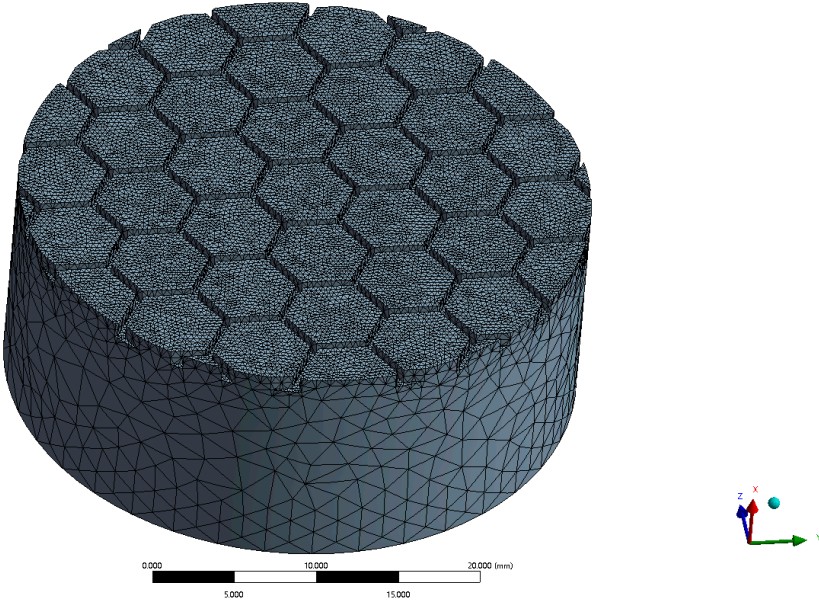

**Figure 9.** Meshing of the hexagonal texture specimens.

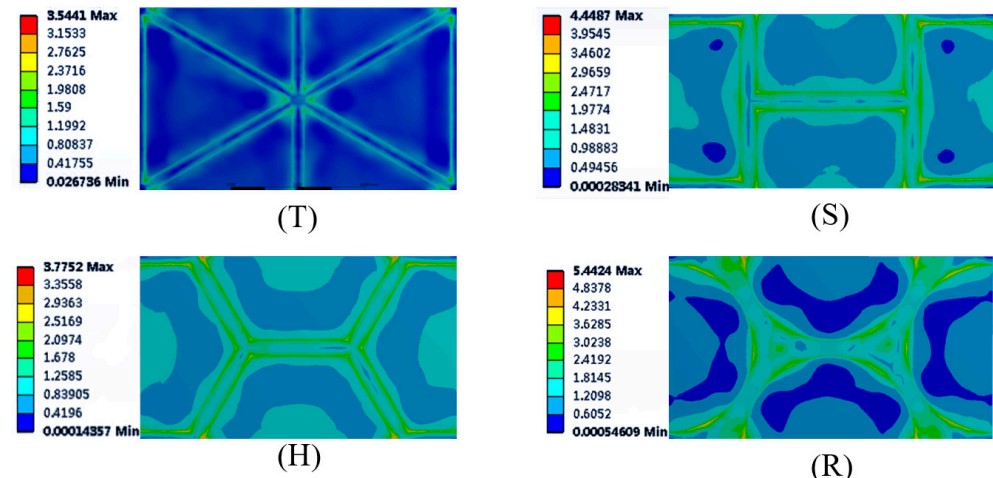

**Figure 10.** The simulated distribution of equivalent stress on the surface of the four cross-grooved texture units. (T) triangle-textured; (S) square-textured; (H) hexagon-textured; (R) round-textured.

There was an interaction of fluid load support and asperity load support that must balance the applied load under mixed lubrication. A high contact stress results in a thinner oil film [43] and reduces the contribution of the fluid load, so that the asperity contact dominates and the friction coefficient increases [39]. In addition, as shown in Figures 4 and 6, although the COF of specimens reduced with the increase in rotating speeds and loads, the four types of textured specimens had less differences in COF with the increase in loads, which is consistent with the experimental results of Zhong et al. [33]. This trend indicates that the asperity contact has a more prominent effect on the lubrication than oil film. In conclusion, the tribological properties of the hexagon-textured specimens were significantly improved compared with those of the other textured specimens at the load of 20 N and the rotating speed of 250 rpm.

## 4. Conclusions

In conclusion, we systematically explored the effects of cross-grooved texture shapes (i.e., triangle, square, hexagon, round) on tribological performance with the change in loads and rotating speeds under mixed lubrication. Overall, the hexagon-textured specimens exhibited lower friction coefficients than the triangle, square, and round-textured specimens under the experimental conditions. More importantly, hexagonal texture has a more prominent effect on lubrication than the three other textures when the loads decreased and the rotating speeds increased.

The reduction in the contact pressure can decrease the acting force of asperity contacts on the surface between specimens and counter-disk, thus lowering the friction force. According to the simulation, the equivalent stress on the hexagon-textured surface was less than square-textured and round-textured, which is beneficial to increase the oil film thickness between the contact surfaces with a hexagon texture. There is an interaction of fluid load support and asperity load support that must balance the applied load under the mixed lubrication. In summary, the asperity contact is of huge importance to load capacity under mixed lubrication, and inevitably affects oil film for lubrication. In our study, the hexagon-textured specimens exhibited better tribological properties under the load of 20 N and the rotating speed of 250 rpm.

**Author Contributions:** Conceptualization, S.H. and L.Z.; Methodology, S.H. and L.Z.; Software, S.H.; Validation, S.H. and L.Z.; Formal analysis, S.H. and L.Z.; Investigation, S.H. and Q.G.; Resources, L.Z.; Data curation, S.H. and L.Z.; Writing—original draft preparation, S.H.; Writing—review and editing, S.H.; Visualization, S.H.; Supervision, L.R., L.Z. and S.H.; Project administration, L.Z.; Funding acquisition, L.Z. All authors have read and agreed to the published version of the manuscript.

**Funding:** This research was funded by the National Natural Science Foundation of China (51905208) and the China Postdoctoral Science Foundation (2020M670855).

**Institutional Review Board Statement:** Not applicable.

**Informed Consent Statement:** Not applicable.

**Data Availability Statement:** Not applicable.

**Conflicts of Interest:** The authors declare no conflict of interest. The funders had no role in the design of the study; in the collection, analyses, or interpretation of data; in the writing of the manuscript, or in the decision to publish the results.

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
