# Peer review of "Influence of Cross-Grooved Texture Shape on Tribological Performance under Mixed Lubrication"

_coatings, doi:10.3390/coatings12030305_

Round 1

Reviewer 1 Report

The paper is indeed well structured and describes novelty clearly. However, it would have been great to include some information regarding the practical reason that has been adopted in the selection of operational parameters? I mean the contact load/speed, temperature, lubricant etc. Are these parameters are related to any practical engineering application? Why these certain laser texture styles were only used in this study? Will it be possible to include some characterisation of tribofilm formed on the surface? 

Author Response

We thank the reviewer for the recognition of our work, as well as the valuable comments. Our reply is as follows:

The selection of these parameters is related to the actual engineering application. The parameters given in this paper are in line with the actual engineering application, such as the working environment of the plunger of the oil well pump (the subject we have already studied). Therefore, these experimental conclusions can be used as theoretical support for practical applications.

The purpose of using these laser texture styles was to investigate the effect of continuous texture and shape on the tribological properties of surfaces.

Some characterisation of the tribofilm have not been studied yet, but this will be the content of our future research.

Reviewer 2 Report

  1. Typos; all the text, tables and captions should be revised for typo mistakes.
  2. English of the text should be revised properly.
  3. There should be a deeper literature survey on texture effecting tribological performances of materials. I recommend to the authors that they can add a table summarizing similar studies and their key findings on this related topic.
  4. Experimental results should be discussed in a schematic figure can be added in the end of the results section to emphasize the importance of the texture on tribology.
  5. Although there is not a separate discussion section, the experimental results should be backed up with scientific literature, otherwise the presented paper will be a data sheet or report only.

Author Response

We thank the reviewer for the recognition of our work, your comments are very helpful, and we have responded to your questions point by point.

Comments 1: Typos; all the text, tables and captions should be revised for typo mistakes. English of the text should be revised properly.

Replay 1

We apologize for the language problems in the original manuscript. The language presentation was improved with assistance from a native English speaker with appropriate research background.

Comments 2: There should be a deeper literature survey on texture effecting tribological performances of materials. I recommend to the authors that they can add a table summarizing similar studies and their key findings on this related topic.

Replay 2

The author has added a table to the text that summarizes the use of textures in practical engineering.

Comments 3: Experimental results should be discussed in a schematic figure can be added in the end of the results section to emphasize the importance of the texture on tribology.

Replay 3

Figures 4 to 7 analyze the tribological properties of the smooth and textured surfaces, and the experimental results show that the hexagonal texture has a lower friction coefficient. Then, the stress distribution of the textured surface was studied by using the simulation software, and the results also showed that the stress distribution of the hexagonal textured surface was better, which was very helpful for the formation of the lubricating film.

Comments 4: Although there is not a separate discussion section, the experimental results should be backed up with scientific literature, otherwise the presented paper will be a data sheet or report only.

Replay 4

The experimental results are backed up with scientific literature. The effect of geometry and working conditions on the tribological properties of hexagonal textures was investigated by Zheng et al. [1] through experiments and numerical simulations, and the experimental results were in general agreement with the simulated stress distribution results. Zhong et al. [2] also investigated the tribological properties of hexagonal textures and found that the friction coefficient decreases with increasing speed and load, similar to the findings in this paper.

[1] Zheng, L.; Gao, Y.; Zhong, Y.; Lu, G.; Liu, Z.; Ren, L. The size effect of hexagonal texture on tribological properties under mixed lubrication. Industrial Lubrication and Tribology 2018, 70.

[2] Zhong, Y.; Zheng, L.; Gao, Y.; Liu, Z. Numerical simulation and experimental investigation of tribological performance on bionic hexagonal textured surface. Tribology International 2019, 129, 151-161.

Reviewer 3 Report

With the title of “Influence of cross-grooved texture shape on tribological performance under mixed lubrication” the authors shown a study about the effect of different surface textures on the improvement in wear resistance/behaviour.

  1. It is not clear the number of specimens used for each corresponding sample. I assume that the authors have used more than one specimen for each sample (as usual), in that case it is necessary to include information about the dispersion of results.
  2. Between lines 100-101 the authors write about a “traditional heat treatment”. In my opinion this item should be described better.
  3. From line 102 one can read that the specimens are in a quenched condition. Why are the specimens studied in a quenched condition? I ask this question because it is very rare to find a real structural steel in a quenched condition (without tempering for example).
  4. Figure 3: I think that the figure it is not agree with the previous paragraph. I believe that the counter disc should be “on/above” the specimen and not “on the side” of the specimen (as it appears in the figure). Sorry if I am wrong!
  5. Figure 9: I assume that the results showed in such figure are from numerical simulation by Finite Element Method (FEM). There is not information about how the numerical simulation by FEM was performed. The distribution of equivalent stress on the surface of four cross-grooved texture units seems to be a key question into the work (as one can read in the conclusions point), so it is necessary an extra information about the topic.

Kind regards.

Author Response

We thank you for the recognition of our work, your comments are very helpful, and we have responded to your questions point by point.

Comments 1: It is not clear the number of specimens used for each corresponding sample. I assume that the authors have used more than one specimen for each sample (as usual), in that case it is necessary to include information about the dispersion of results.

Replay 1

Does the information about the dispersion of results refer to experimental error? If so, then the experimental results in this paper already contain information about the experimental error, as shown in Figures 4 and 6.

Comments 2: Between lines 100-101 the authors write about a “traditional heat treatment”. In my opinion this item should be described better.

Replay 2

The sequence of the heat treatment process for AISI 1045 steel is annealing, quenching and tempering. We are sorry that some other details of the heat treatment process are not known to us at this time and therefore cannot be described in detail here.

Comments 3: From line 102 one can read that the specimens are in a quenched condition. Why are the specimens studied in a quenched condition? I ask this question because it is very rare to find a real structural steel in a quenched condition (without tempering for example).

Replay 3

Thank you for pointing this out, as you said, the sample is rarely found in the real structural steel in the quenched state and must be tempered.

Comments 4: Figure 3: I think that the figure it is not agree with the previous paragraph. I believe that the counter disc should be “on/above” the specimen and not “on the side” of the specimen (as it appears in the figure). Sorry if I am wrong!

Replay 4

As you think, the counter disc is on/above the specimen. As described on lines 133-134.

Comments 5: Figure 9: I assume that the results showed in such figure are from numerical simulation by Finite Element Method (FEM). There is not information about how the numerical simulation by FEM was performed. The distribution of equivalent stress on the surface of four cross-grooved texture units seems to be a key question into the work (as one can read in the conclusions point), so it is necessary an extra information about the topic.

Replay 5

Thanks for your suggestion, we have supplemented the text with information about numerical simulations.

The simulation experiment in this part is completed in ANSYS Workbench.

The conclusions about the equivalent stress distribution are supplemented in Line 241-243, “A high contact stress results in a thinner oil film and reduces the contribution of the fluid load, so that the asperity contact dominates and the friction coefficient increases”.

Reviewer 4 Report

Journal:  Coatings

Ref manuscript ID coatings-1510186

Manuscript title “Influence of cross-grooved texture shape on tribological performance under mixed lubrication”

Comments to authors:

Comments to authors:

I have carefully reviewed and I do not recommend the publication of the above-mentioned article in  Coatings because the technical merit, significance, Clarity of the objectives and the research question, Clarity and explanation of disciplinary references, Quality of data production, Quality of data analysis, Presentation of results and discussion, quality of the contribution presented by the authors and originality of the work is unclear to the reader .

  • I found that the article does not present any new information, it seems to me that the figures and all equations of the document are copied from a technical book, compared to other research works in the field, there is indeed no clear methodology or analysis are provided here.

the originality is very low here, even the experimental procedure has a very weak argument and is not clear here

The method employed by the authors is very popular using, the content of the article is almost similar to those already published by other authors, the same model , the same figures et  , even - cross-grooved texture shape is well known and can be found in several articles,,,, I cite as an example here,

  • Study on tribological performance of groove-textured bioimplants
  • Effect of geometrical parameters in micro-grooved crosshatch pattern under lubricated sliding friction
  • Influence of Groove Dimensions on the Tribological Behavior of Textured Cylindrical Roller Thrust Bearings under Starved Lubrication
  • Tribological behavior of grooves textured thrust cylindrical roller bearings under dry wear
  • Multi-Scale Surface Texturing in Tribology—Current Knowledge and Future Perspectives
  • Effect of Low Depth Surface Texturing on Friction Reduction in Lubricated Sliding Contact
  • Tribological performance of surface texturing in mechanical applications—a review
  • Effect of grooved surface texturing on the behavior of lubricated contacts
  • The entire document appears messy and requires extensive improvement, generally improving its overall presentation to properly achieve the purpose of the scientific contribution. Any document must have a coherent structure. But, I don't see this in this document
  • Rejected the paper due to lack of any sound analysis, measurement data and that the central idea makes less sense.
    The authors are suggested to take the reviews seriously into consideration and do a major overhaul of the manuscript and if they like to then submit it as a new manuscript.
  • Moreover, the English language used in the paper is quite poor, and does not meet the standard requirements of the journal
  • There is no new problem solved in this paper and no new information is produced. it is very poor contains only figures or drawings to be found in technical books didn't offer any new information.

According to my recommendations which are very useful to authors, this work is not able to accept for publication.

Therefore , for the said reasons , I do not recommend the publication of this document final conclusion  (Reject )

Thank you for inviting to review this document

Best Regard

Author Response

We are grateful to the reviewer for pointing out these issues.

Experimental and simulation studies on continuous groove textures with four shapes are carried out in this paper, and the results show that the hexagonal texture has better tribological properties. The results are very similar to the conclusions drawn by other literatures. The content of this study is to highlight the superiority of the hexagonal structure, which is very common in nature (e.g., honeycomb, tree frog’s feet, etc.). The author believes that the texture of hexagons is very worthwhile to study.

Round 2

Reviewer 2 Report

The required revision are made.

Author Response

Thank you for taking the time to review my manuscript!

Reviewer 3 Report

Dear authors, you can se my commenst in the attached file.

Best regards!

Author Response

Thanks for your valuable suggestions and comments, we have completed the correction according to your request.

  1. The number of samples used for each corresponding sample is indicated. Five specimens were repeated for each condition to reduce test errors.
  2. We have redrawn the schematic diagram of the frictional test equipment, as shown in Fig 3.
  3. Information on FEM numerical simulations is supplemented, between lines 227-232. As follows

    “Apply a series of loads (20, 50, 70 and 100 N) to the textured surface. The mesh was generated by tetrahedral elements, the element size is 3 mm. The mesh on the surface of the tested specimen is refined with the element size is 0.2mm. Figure 9 shows the meshed of hexagonal texture specimen. Other options use default values. By applying the analysis over the texture surface under load we get the stress distribution in numerical”

Round 3

Reviewer 3 Report

Dear authors.

Thank you for your kind reply. In my opinion the thrid version of the work is better than the first one.

Congratulations and king regards!